# Metabolic Effects of Brown Adipose Tissue Activity Due to Cold Exposure in Humans: A Systematic Review and Meta-Analysis of RCTs and Non-RCTs

**DOI:** 10.3390/biomedicines12030537

**Published:** 2024-02-28

**Authors:** Shirin Tabei, Rodrigo Chamorro, Sebastian M. Meyhöfer, Britta Wilms

**Affiliations:** 1Institute of Endocrinology and Diabetes, University of Lübeck, 23562 Lübeck, Germany; shirin.tabei@uni-luebeck.de (S.T.); rchamorro@uchile.cl (R.C.); sebastian.meyhoefer@uni-luebeck.de (S.M.M.); 2Center of Brain, Behavior, and Metabolism (CBBM), University of Lübeck, 23562 Lübeck, Germany; 3German Center for Diabetes Research (DZD), Partner Düsseldorf, 85764 München-Neuherberg, Germany; 4Department of Nutrition, Faculty of Medicine, University of Chile, Santiago 8380453, Chile

**Keywords:** brown adipose tissue, cold exposure, glucose metabolism, lipid metabolism, humans

## Abstract

Brown adipose tissue (BAT), specialized in thermoregulation in mammals, has been linked to improved glucose and lipid homeostasis when activated by cold exposure (CE). This systematic review and meta-analysis assessed the metabolic effects of CE-induced BAT activation in healthy humans, examining changes in glucose and lipid metabolism compared to thermoneutrality (TN). A literature search was conducted, identifying relevant human studies, including randomized controlled trials (RCTs) and non-RCTs, based on predefined inclusion criteria. Seven studies (a total of 85 participants) fully met the criteria. Data on plasma glucose, insulin, triglycerides (TGs), and free fatty acids (FFAs) were extracted for meta-analysis. When comparing TN and CE under fasting conditions, there were no significant changes in glucose, insulin, or TG concentrations (all *p* > 0.36). In contrast, CE significantly increased FFA concentrations (*p* = 0.002; n = 38). Bias was absent for all parameters, but heterogeneity was observed for insulin (I^2^ = 74.8%). CE primarily affects FFA concentration, likely reflecting cold-induced BAT activity. This suggests that circulating FFAs, serving as the primary fuel for thermogenesis, could indicate BAT activation. However, understanding the effects of BAT activation on overall metabolism requires a broader approach beyond fasting glucose and lipid concentration measurements.

## 1. Introduction

Obesity has reached epidemic proportions [1]. Obesity is associated with metabolic diseases such as type 2 diabetes (T2D), dyslipidemia, or metabolic syndrome. In terms of thermodynamics, if energy intake exceeds energy expenditure, a positive energy balance results in subsequent body weight and fat mass gain and, if persistent, in the development of obesity. However, a low success rate of weight reduction by decreasing caloric intake has been reported [2]. Long-term maintenance of weight loss is difficult and may ultimately lead to weight gain after an initial phase of weight loss [3,4].

Several animal and human studies have addressed strategies to increase energy expenditure to force a negative energy balance with subsequent weight loss, for example, by increasing physical activity [5,6].

In recent years, and after the rediscovery of brown adipose tissue (BAT) in adult humans [7,8,9], activation and recruitment of this fat tissue moved to the center of scientific interest as a potential anti-obesity and anti-diabetes target.

BAT is a highly specialized type of fat tissue mainly found in newborns of mammals and human infants, whose primary functions involve non-shivering thermogenesis (NST) [10] and the production of heat to maintain normal body temperature [11]. Breastfeeding plays an important role in the development of thermogenic fat in human newborns [12,13,14]. After birth, maternal cues including free fatty acids (FFAs) and alkylglycerols are delivered through milk [12,15] and microbial products of the gut microbiota to infants [16], which triggers thermogenic (beige/brown) fat development. The mechanism behind this development is due to beta-adrenergic signaling [15].

In adults, BAT is most often located in the supraclavicular region and, to a lesser extent, in the abdomen, around the spine, and around the heart and kidney [17]. Both experimental and clinical studies have shown that activated BAT can potentially enhance glucose metabolism [18] and lipid metabolism [19], increase metabolic rate [20], and promote the loss of fat mass in humans [21]. Interestingly, preserving body core temperature in adults depends on the subcutaneous white adipose tissue and its insulating function rather than thermogenic fat [22].

Researchers have investigated different mediators to activate BAT, including diet, substrates (e.g., capsaicin), medications such as beta-adrenergic (β-AR) agonists (e.g., mirabegron), and cold exposure (CE) [8,23,24,25]. CE has been reported as the most effective activator of BAT [26]. Mild CE stimulates the activity of the sympathetic nervous system, leading to the release of norepinephrine (NE). NE then binds to β-AR receptors located on the membrane of brown adipocytes [27]. Furthermore, intracellular signaling cascades lead to triglyceride (TG) degradation into FFA, which interacts with uncoupling protein 1 (UCP1). UCP1 is a hallmark of BAT [28] and is highly abundant on the inner mitochondrial membrane. If activated, UCP1 uncouples the electron transport chain independently of the consumption of adenosine triphosphate, leading to a free flow of protons across the inner mitochondrial membrane. This process, in turn, leads to NST and the production of heat [11]. By comparing Positron Emission Tomography and Computed Tomography (PET/CT) images before and after 2 h of CE at 19 °C, [^18^F]-Fluorodeoxyglucose ([^18^F]-FDG) uptake in BAT is increased, specifically in the supraclavicular region, indicating the potential of mild CE to activate BAT [8,9].

Studies in rodents have provided evidence of improved insulin sensitivity and browning in white adipose tissue in response to CE in obese mice [29,30,31]. Bartlett and colleagues showed a TG clearance after 24 h of CE in hypertriglyceridemic mice, specifically mediated by BAT [32]. In humans, Blondin et al. reported a 45% increased BAT oxidative metabolism and BAT volume in healthy young lean and overweight men by daily CE at 10 °C, who experienced low shivering for 2 h over 4 weeks [33]. We have also shown that 2 h of moderate CE at 18 °C increased BAT activity and improved insulin sensitivity in healthy lean participants [34].

In sum, substantial evidence suggests that CE leads to increased BAT activity and subsequent beneficial metabolic adaptations in humans.

This systematic review and meta-analysis comprehensively gather and evaluate the impact of CE without experience of shivering on healthy humans compared to TN. Specifically, it focuses on glucose and lipid metabolism only at the fasting level to ascertain whether mild CE influences the aforementioned parameters at a basal level.

## 2. Methods

### 2.1. Search Strategy

Computerized searches were performed in PubMed (MEDLINE) and Web of Science, respectively, with the initial search conducted on 7 March 2022, and the latest search performed on 29 March 2023. The following search terms were used:


*“(((“Brown Adipose Tissue”“ OR ““BAT”“ OR ““Brown Fat”“) AND “(“Cold Exposure”“ OR ““Cold Expos*”“ OR ““Cold Stimulus”“ OR ““Cold Induced”“ OR ““Cold Acclimation”“ OR ““Cold Effect”“) AND “(“Metabolic Homeostasis”“ OR ““Metabolic Effect”“ OR ““Metabolic Response”“ OR ““Metabolic Consequence”“ OR ““Metabolic Regulation”“ OR ““Metabolic Impact”“ OR ““Metabolism”“) AND “(“Humans”“ OR ““Subjects”“ OR ““Participants”“ OR ““Individuals”“)) NOT “(“Animals”“)).*


### 2.2. Inclusion and Exclusion Criteria

The protocols of studies investigating the effects of CE on BAT activity vary in design (i.e., comparing CE vs. TN), the temperature level of the CE (i.e., from 4 °C to 21 °C), and regarding cooling procedures (i.e., using cooling feet and hands or water-perfused suits), and the duration of CE. Some studies have used cooling protocols that allowed low shivering [33,35,36,37], whereas others avoided shivering (see Section 3.1). Therefore, to extract data for this meta-analysis, we aimed to select studies that are as homogenous and comparable as possible (Table 1 and Table 2). We included original studies, RCTs, and non-RCTs conducted in healthy adults (≥18 years of age), both male and female, and of any geographic area. Studies were included if the temperature range of intervention was between 12–19 °C and between 20–25 °C for CE and TN conditions, respectively. The range of temperature refers to the general average of previous applied cooling studies in the literature. 

Additionally, studies with personalized cooling protocols, in which the CE temperature was set above the shivering threshold of each subject individually, were also included. Lastly, concerning metabolic outcome parameters, research was restricted to studies reporting basal glucose, insulin, TG, and FFA concentrations. Studies with either lower (<12 °C) or higher (>26 °C) temperatures were considered “not mild CE” or “warm but not TN” and were, therefore, excluded. Studies reporting a shivering observation during the CE protocol used and studies with shorter (i.e., nighttime fasting lower than 8 h) or longer fasting periods (i.e., 24 h or more) were also excluded. 

### 2.3. Data Selection

Following the Preferred Reporting Items for Systematic Reviews and Meta-Analysis (PRISMA) guidelines [45], three investigators (ST, RC, and BW) independently screened all titles and abstracts of papers to assess them for potential relevance. If no agreement was achieved, the respective titles were reviewed by a fourth researcher (SMM). The systematic review and meta-analysis included studies conducted on healthy adults who underwent both TN and CE and fulfilled all inclusion and exclusion criteria (see Section 2.2).

### 2.4. Meta-Analytic Approach

The meta-analysis procedure was performed using the RStudio (version 2023.06.02, Boston, MA, USA) library package “metafor” (version 2.0-0). Means and standard deviations (SDs) of fasting glucose, insulin, TG, and FFA concentrations during both CE and TN were extracted (Table 2).

If needed, a 95% confidence interval (CI) was converted to an SD using the following equation:$$\mathrm{SD}\;=\sqrt{N}\times \left(\mathrm{upper}\;\mathrm{limit}\;-\;\mathrm{lower}\;\mathrm{limit}\right)/4.128$$

A random-effects model was used to measure the impact of TN vs. CE on glucose, insulin, TG, and FFA concentrations. The effect size was provided by the standardized mean difference (SMD), calculated as follows: SMD = new treatment improvement − comparator (placebo) improvement/pooled standard deviation

The studies’ 95% CI and corresponding Z values (i.e., ±1.96) were estimated. The findings of the meta-analysis were graphically displayed by forest plots. When the 95% CI from a single study (or the pooled estimate) crosses the line of no effect, the difference in outcome between CE and TN is not statistically significant. Heterogeneity was assessed with Cochran’s Q test and I^2^ statistics. Cochran’s Q test is calculated as the weighted sum of squared differences between individual study effects and the pooled effect across studies. The I^2^ statistic describes the variability in effect estimates due to heterogeneity rather than sampling error. The following formula was used to quantify the degree of inconsistency or heterogeneity among the individual studies included in the analysis (df: degrees of freedom; Q: chi-squared statistics) [46]:I^2^ = (Q − df/Q) × 100

Conventions were followed regarding the interpretation of I^2^: 0–40% may represent no heterogeneity; 30–60% may represent moderate heterogeneity; 50–90% may represent substantial heterogeneity, and 75–100% may represent considerable heterogeneity [47]. Publication bias was assessed by funnel plots, where the standard error was plotted on the vertical axis with a reversed scale, placing larger studies towards the top, and effect estimates were given on the horizontal axis [48]. In addition, Egger tests, in which the standardized effect sizes were regressed on their standard error, were applied. Within this test, the regression intercept is expected to be zero in the absence of publication bias. *p* values lower than 0.05 were considered statistically significant.

## 3. Results

### 3.1. Study Selection and Characteristics

Literature research was performed in March 2023 in Web of Science and PubMed (MEDLINE) databases and identified 844 potentially relevant articles (Web of Science n = 358; PubMed n = 486). In addition, 30 articles were included after reviewing the references of identified articles. After removing duplicates, articles were assessed for more detailed evaluation. Following the selection process, 173 articles were assessed for eligibility. After reading full texts, 151 articles were removed, and ten studies fulfilled the protocol criteria (TN and CE conditions). Seven studies provided data on at least one of the selected metabolic parameters and were included in the meta-analysis. The PRISMA flow chart is shown in Figure 1.

### 3.2. Systematic Review and Meta-Analysis

Based on the predefined criteria, seven studies were included in this meta-analysis [38,39,40,41,42,43,44]. Details regarding these studies and the extracted raw data are presented in Table 1 and Table 2.

Plasma glucose and TG values were converted into mmol/L, insulin values were converted into pmol/L, and plasma FFA values were converted into µmol/L.

### 3.3. Exclusion of Parameters after Conversion

Wijers and colleagues presented the insulin concentration in nM [39], and converting it into pmol/L (by multiplying by 1000) resulted in unphysiologically high values. Thus, these values were not considered for the meta-analysis. Also, the conversion of insulin values in the study of Vosselman et al. revealed unphysiological concentrations, and they were not included in the meta-analysis [40].

### 3.4. Characteristics of Participants

A pooled sample size of n = 85 healthy adults was included in the analysis. The sample size of respective studies ranged from 7 to 18 participants. The mean age of participants across the studies was 31.8 ± 11.0 years, with a mean BMI of 24.9 ± 3.4 kg/m^2^. The mean temperature under the TN condition was 24.0 ± 1.9 °C, and under the CE condition 17.1 ± 1.1 °C, respectively (Table 1). All studies reported a non-shivering observation using hourly questionnaires, personalized cooling, self-reporting, or electromyography (EMG). The experimental visits were performed after an overnight fast.

### 3.5. Meta-Analytic Results

The average data of glucose, insulin, TG, and FFA concentrations obtained in enrolled studies are provided in Table 2.

#### 3.5.1. Effects of CE on Plasma Glucose

Four studies are included (k = 4) with a pooled sample size of n = 38. For all studies, the 95% CI intersects the line of no effect size. The pooled effect size of glucose was 0.19 (SMD), and results indicated no effect of CE in comparison to TN (95% CI = [−0.27, 0.65], *p* = 0.41). The test of heterogeneity was not significant (Q = 2.8, *p* = 0.43, I^2^ = 2.3%) (Figure 2A). Visual inspection of the funnel plot and the result of the Egger intercept (z = 0.59, *p* = 0.56) suggest the absence of publication bias (Figure 2B).

#### 3.5.2. Effects of CE on Plasma Insulin

Four studies are included (k = 4) assessing the effects of CE as compared to TN on fasting plasma insulin concentrations, with a pooled sample size of n = 50. The pooled effect size of insulin concentration was 0.39 (SMD), and results indicated no significant changes in insulin concentration post-CE compared to TN (95% CI = [−0.44, 0.12], *p* = 0.36). Q and I^2^ tests indicated the presence of statistical heterogeneity (Q = 11.53, *p* = 0.009; I^2^ = 74.8%) (Figure 3A). No publication bias existed by visual inspection of the funnel plot and interpretation of the Egger intercept (z = 0.39, *p* = 0.69) (Figure 3B).

#### 3.5.3. Effects of CE on Plasma Triglyceride

Four studies are included (k = 4) with a pooled sample size of n = 45. The pooled effect size was −0.17 (SMD), and the results indicated no differences between TN and CE (95% CI = [−0.58, 0.25], *p* = 0.43). Heterogeneity was not evident according to Q and I^2^ statistics (Q = 3, *p* = 0.68, I^2^ = 0.0%) (Figure 4A). Visual inspection of the funnel plot and the Egger intercept (z = −1.09, *p* = 0.28) negates a publication bias (Figure 4B).

#### 3.5.4. Effects of CE on Plasma Free Fatty Acids

Three studies (k = 3) with a pooled sample size of n = 38 reported data on plasma FFA concentration. The pooled effect size was −0.75 (SMD), and the results indicated a significant increase in FFA concentration post-CE compared to TN (95% CI = [−1.21, −0.28], *p =* 0.002). Further analysis revealed no heterogeneity (Q = 2, *p* = 0.67, I^2^ = 0.0%) (Figure 5A) as well as no publication bias by visual inspection of the funnel plot and by the Egger intercept (z = −0.39, *p* = 0.70) (Figure 5B).

## 4. Discussion

The current systematic review and meta-analysis aimed to analyze the data of glucose and lipid parameters and analyze the effect of mild cold only at the fasting level. Prior research has established that mild CE improves glucose and lipid metabolism in humans and rodents. Generally, a high volume of active BAT is associated with lower glycated hemoglobin (HbA1c), and improved diabetes status [31]. In this scenario, an increased expression of glucose transporter type 4 in brown adipocytes post-CE was reported in a mouse model, which was associated with increased glucose clearance [49,50]. Comparable effects of improved insulin sensitivity in humans have been confirmed.

Interestingly, one of the main findings of the present meta-analysis showed that basal glucose concentration was not changed by mild CE as compared to TN. However, subsequent animal and human studies reported improvements in glucose homeostasis parameters, such as improved insulin sensitivity and increased glucose infusion by performing the Botnia Clamp technique after short- and long-term CE in both healthy individuals and patients with T2D [51]. However, this did not lead to any significant changes in fasting plasma glucose. The Botnia Clamp method enables the detection of a more comprehensive picture of glucose metabolism [34,52]. In a large study including 260 healthy and lean participants, after 6–12 h of fasting, they underwent 2 h of mild CE at 19 °C, which did not change fasting plasma glucose and insulin levels compared to TN (*p* = 0.07 and *p* = 0.47, respectively). However, in the same study, HbA1c showed a negative association with BAT volume [53]. Similarly, no glucose and insulin level differences were observed between CE and TN after 4 h of fasting [54]. In mice, 14 days of CE improved glucose tolerance and insulin sensitivity but not basal glucose levels [55]. Additionally, CE combined with a high-fat diet for 6 weeks improved insulin sensitivity and decreased plasma glucose levels during an oral glucose tolerance test (OGTT) in mice, whereas the fasting blood glucose level again remained unchanged [56]. Lebbe et al. reported an increased plasma glucose level in cold-housed mice compared to the mice housed at room temperature; however, the BAT glucose uptake increased after cold exposure [57]. Taken together, these findings emphasize that fasting glucose levels may not capture the nuanced changes in glucose homeostasis induced by CE, thereby necessitating the use of gold-standard methods to assess its true impact on metabolism. Additionally, different results between human and rodent studies can be explained by the physiological differences and applied cooling protocols.

Multiple different variables can impact BAT activity. For instance, BMI negatively correlates with BAT volume and activity after CE [58]. However, it could be shown that CE, compared to TN, did not improve fasting glucose levels in healthy young subjects with either normal weight, overweight, or obesity [59]. This systematic review incorporated studies with a varied BMI range, encompassing participants categorized as lean (n = 4) and overweight (n = 3) and with an overall mean BMI of 24 kg/m^2^. According to existing literature, this mean BMI suggests the inclusion of participants exhibiting active BAT volume. 

In contrast, a study, after dividing the cohort based on the presence of BAT into positive and negative subgroups, could show a decreased fasting glucose plasma level after CE compared to TN in only the BAT+ subgroup [60]. The research strategy of this work was restricted to only healthy individuals who were exposed to almost the same cooling temperature irrespective of their anthropometric or environmental parameters, which may affect the changes in fasting glucose levels. Furthermore, assessment of only fasting glucose concentration might not provide a reliable indicator of changes in glucose metabolism, i.e., due to cold [61,62].

At the molecular level, activation of the sympathetic nervous system and epinephrine secretion due to CE can inhibit pancreatic insulin secretion. Studies in rats have shown decreased insulin secretion post-CE [63,64,65]. In contrast, inhibition of the sympathetic and activation of the parasympathetic nervous systems in rats stimulate β-cell proliferation [66]. In cold-induced BAT activity, enhancing the insulin receptor signaling pathway appears to be a crucial mechanism through which BAT optimizes its glucose uptake efficiency after 4 h at 4 °C [63]. In contrast to animal studies, this meta-analysis found no changes in fasting insulin levels after CE. This is supported by the study from Iwen and colleagues that did not report any changes in peripheral fasting insulin secretion or first-phase insulin response post-CE compared to TN. However, whole-body insulin sensitivity was improved [34]. Generally, insulin concentration in serum is a sensitive parameter, which can be easily affected by others, such as meals before the measurement. Remarkably, the liver extracts 50% of postprandially secreted insulin in healthy individuals before entering the systemic circulation.

Additionally, serum insulin is an unstable molecule with a short half-life. Therefore, in the clinic, C-peptide is often a more reliable alternative to insulin for assessing β-cell activity and diagnosing diabetes status since C-peptide is generally found in the bloodstream in approximately equal proportions to insulin [67]. As previously mentioned, to ensure a more homogeneous cohort concerning BAT prevalence and activity, it is possible to divide the cohort into BAT+ and BAT− subgroups based on cold-induced glucose uptake in the supraclavicular region assessed by PET/CT. Crandall et al. showed a decrease in insulin levels, specifically in the BAT+ group, but not in the BAT− group after 2 h of CE. Another study that compared BAT+ and BAT− individuals following 2 h of CE observed a significant reduction in insulin levels in the BAT+ subgroup [68]. These data point to the necessity to take a closer look at individual BAT activity to elucidate the effects of CE on metabolism in humans.

Regarding the impact of CE on FFAs, findings supported the observation of elevated FFA levels following CE compared to TN. This increase in FFAs was consistent across all studies included in this meta-analysis. Brown adipocytes utilize fatty acids released from white adipose tissue for adaptive thermogenesis [32]. Furthermore, an association exists between BAT volume and elevated lipolysis in mice due to activation with β-ARs agonist [69], which stimulates lipolysis and increases FFA levels. By comparing individuals with detectable vs. undetectable BAT, it has been shown that the BAT+ group exhibits higher rates of FFA oxidation during CE than the BAT− group [70]. 

In hyperlipidemic Apoa5-/- mice, it was demonstrated that BAT, only after 4 and later 24 h of CE at 4 °C compared to mice housed at room temperature, effectively extracts a substantial amount of TG from circulation due to the high oxidative capacity of activated BAT [32], which underpins the role of TG as fuel for NST due to CE. However, human studies did not find a reduction in circulatory TG post-CE. Indirect evidence from human studies reported that intracellular TG is the primary fuel of BAT thermogenesis [71]. In rats, following both acute and chronic CE at the temperature of 10 °C for 6 h and 21 days, respectively, the inhibition of TG using nicotinic acid (NiAc) results in reduced iBAT oxidative metabolism, manifested by decreased glucose and lipid uptakes [57]. Inhibition of TG lipolysis by NiAc application suppressed cold-induced thermogenesis in healthy humans [72]. Experimental studies in humans have reported either no changes or an increase in TG levels during CE [35,36,37,72,73]. In contrast, Iwen and colleagues showed a significantly decreased TG level after 2 h of acute mild CE compared to TN [34]. In a study comparing BAT+ and BAT− groups after an acute CE before PET/CET imaging, no differences in TG concentration were observed between the two subgroups.

When evaluating BAT activity, PET/CT counts as a gold standard. However, its limitations are due to high radiation exposure and its invasive nature. For infants and children, infrared thermography offers an alternative, non-invasive, and indirect method, providing initial insights into the presence of thermogenic tissue in the supraclavicular region. Nevertheless, its application in this demographic is restricted by the requirement for individuals to remain calm and motionless during the assessment.

From a conceptual point of view, the application of an oral fat tolerance test (OFTT) has been known to assess intestinal fat metabolism in rodents and humans by consumption of high-fat meals [74,75], which could provide a more comprehensive insight into the hypothesized effects of CE on lipid metabolism.

### Limitations

The main limitation of the present systematic review and meta-analysis was the inclusion of both RCT and non-RCT studies. This decision was made based on the limited number of human studies that met all the predetermined inclusion and exclusion criteria.

Studying the metabolic effects of BAT activity in humans is complex due to their susceptibility to different variables such as dietary intake and physical activity days before measurements, the metabolic health of participants, and BAT activation and volume.

Furthermore, although gender is an important factor in any metabolic analysis such as BAT activity [76], the studies incorporated in this systematic review predominantly featured male participants, which counts as one of the limitations of this systematic review.

Moreover, age constitutes another parameter influencing metabolic outcomes, including a negative correlation with BAT activity [76]. Within this systematic review, five out of seven studies involved participants aged between 20–30 years, while only two studies included individuals with a mean age of 47 years old. Due to the limited sample size, we refrained from dividing the included studies based on age for our analysis.

Moreover, the cooling protocols employed across various studies stand out as the pivotal determinant influencing outcomes concerning the impact of CE on BAT activity [77]. Therefore, in this systematic review, we defined rigorous criteria based on cooling temperature for the study selection, which led to a very small number of included studies. However, it is important to mention that the duration of the applied protocols varied between included studies. In future studies, a standardized cooling protocol, addressing both temperature and duration, will be very helpful for comparing the metabolic impact and BAT activity in human studies. 

Additionally, dividing cohorts into BAT+ and BAT− subgroups would be a promising approach to better investigate the effects of CE on metabolism with more nuance. 

Nonetheless, concentrating on these two subgroups would have further reduced the already limited pool size. Hence, it was not incorporated into this meta-analysis.

## 5. Conclusions

This meta-analysis evaluated the CE-induced changes in metabolic parameters under fasting conditions in humans. Taken together, we were able to establish that increased fasting FFAs are likely the predominant substrate fueling BAT oxidative metabolism in humans and could serve as an indicator of BAT activation at the basal level due to CE. FFA concentration was elevated after mild CE in all included studies, which indicates that the body undergoes increased lipolysis and releases more FFAs as an energy source for BAT to maintain the body temperature due to NST. By analyzing the additional parameters, TG, glucose, and insulin, no significant changes at the fasting level have been observed after mild CE compared to TN. Regarding glucose homeostasis, relying only on fasting glucose and insulin levels may not provide sufficient insight into the effect of CE on BAT activity, although the included studies have already provided evidence of activated BAT as increased energy expenditure post-cold exposure. Therefore, for a more comprehensive understanding of the beneficial effects of CE on glucose homeostasis, gold-standard methods such as the Botnia Clamp will be very important, which is the gold standard to assess beta cell function and insulin sensitivity. Considering lipid homeostasis, an OFTT has been known to determine intestinal fat metabolism in rodents and humans by consumption of high-fat meals [75,78], and monitoring blood TG levels could count as a complementary method to the monitoring of fasting FFA and TG. 

Generally, in clinical practice, the assessment of BAT volume in the early stage of obesity is important. It serves as an early indicator, signaling the potential risk of metabolic syndrome associated with obesity in the future. Additionally, dividing cohorts into BAT+ and BAT− subgroups would be a promising approach to an in-depth assessment of the metabolic effects of CE in humans.

## Figures and Tables

**Figure 1 biomedicines-12-00537-f001:**
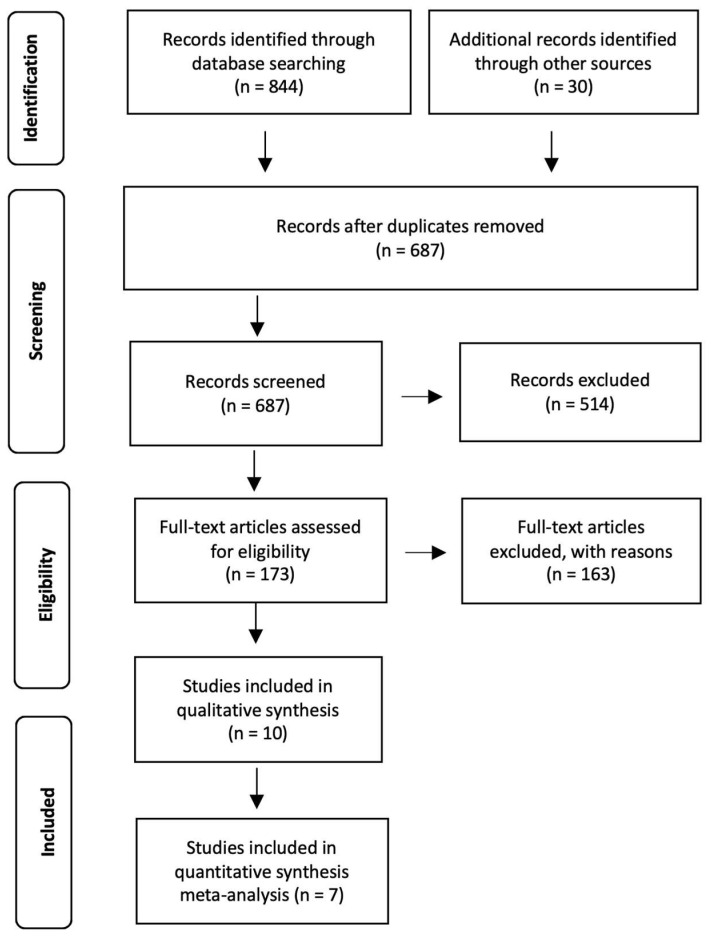
PRISMA 2009 flow diagram of systematic review and meta-analysis.

**Figure 2 biomedicines-12-00537-f002:**
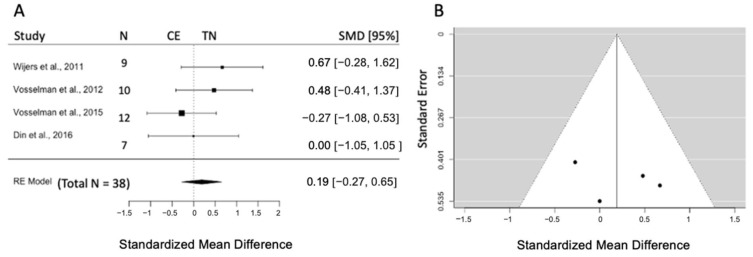
(**A**) Forest plot represents the effect size and 95% CI for plasma glucose concentration post-CE compared to TN. (**B**) Funnel plot checks the existence of publication bias. RE: random effect; SMD: standardized mean difference; CI: confidence interval; CE: cold exposure; TN: thermoneutrality [39,40,41,42].

**Figure 3 biomedicines-12-00537-f003:**
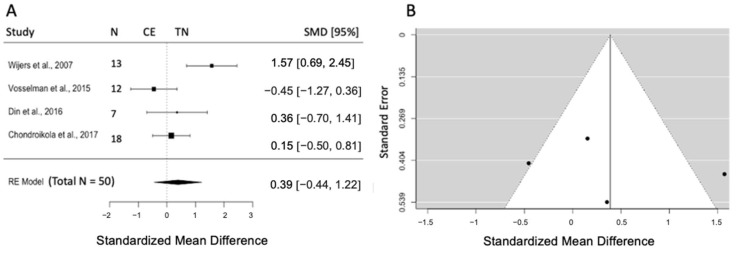
(**A**) Forest plot represents the effect size and 95% CI for plasma insulin concentration post-CE compared to TN. (**B**) Funnel plot to check the publication bias. RE: random effect; SMD: standardized mean difference; CI: confidence interval; CE: cold exposure; TN: thermoneutrality [38,41,42,44].

**Figure 4 biomedicines-12-00537-f004:**
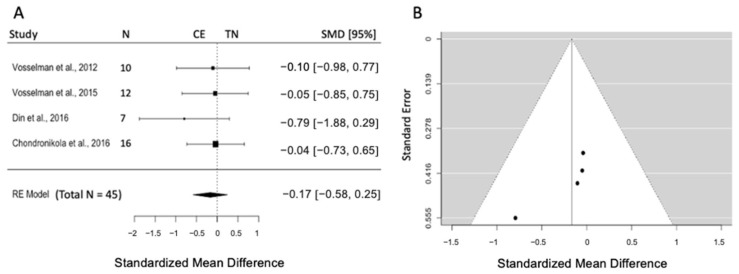
(**A**) Forest plot represents the effect size and 95% CI in plasma TG concentration post-CE compared to TN. (**B**) Funnel plot to check the publication bias. RE: random effect; SMD: standardized mean difference; CI: confidence interval; TG: triglyceride; CE: cold exposure; TN: thermoneutrality [40,41,42,43].

**Figure 5 biomedicines-12-00537-f005:**
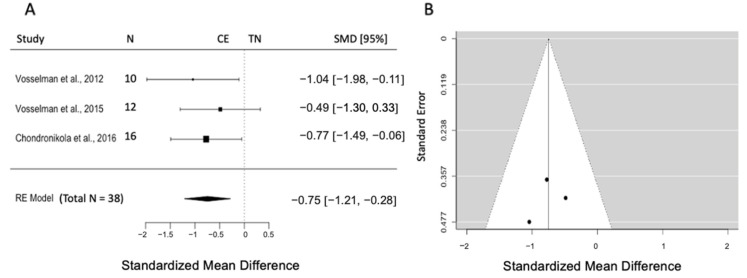
(**A**) Forest plot represents the effect size and 95% CI in plasma FFA concentration post-CE compared to TN. (**B**) Funnel plot to check the publication bias. RE: random effect; SMD: standardized mean difference; CI: confidence interval; FFA: free fatty acid; CE: cold exposure; TN: thermoneutrality [40,41,43].

**Table 1 biomedicines-12-00537-t001:** Summary of the protocol included.

Studies		TN		CE			
Reference	Year	Temperature (°C)	Duration	Temperature (°C)	Duration	Cooling Method	Shivering Test
Wijers et al. [38]	2007	22.0	36 h	16	36 h	Respiration chamber	Hourly questionnaire
Wijers et al. [39]	2011	22.0	36 h	16	48 h	Respiration chamber	Hourly questionnaire
Vosselman et al. [40]	2012	24.5	45 min	PC	120 min	Air conditioner	Non-shivering PC
Vosselman et al. [41]	2015	RT	60 min	PC	120 min	Flow-controlled circulating bath	Non-shivering PC
Din et al. [42]	2016	22.0	120 min	PC	240 min	Air conditioner	Observation and self-report
Chondronikola et al. [43]	2016	26.2	5 h	18.2	5 h	Cooling blanket	Non-shivering PC
Chondronikola et al. [44]	2017	26.2	5 h	18.2	5 h	Cooling blanket	Non-shivering PC
Mean ± SD temperature	-	24.0 ± 1.9 °C	-	17.1 ± 1.1 °C	-	-	-

Abbreviations: room temperature (RT); personalized cooling (PC); cold exposure (CE); thermoneutrality (TN).

**Table 2 biomedicines-12-00537-t002:** Characteristics of studies.

Studies			Participants’ Characteristics				TN				CE			
Reference	Year	Design	n	Male/ Female	Age (yrs.)	BMI (kg/m^2^)	Glucose (mmol/L)	Insulin (pmol/L)	TGs (mmol/L)	FFAs (µmol/L)	Glucose (mmol/L)	Insulin (pmol/L)	TGs (mmol/L)	FFAs (µmol/L)
Wijers et al. [38]	2007	Comparative study	13	male	22.8 ± 1.7	22.9 ± 0.9	NA	101.4 ± 9.6	NA	-	NA	88.3 ± 6.2	NA	-
Wijers et al. [39]	2011	Non-RCT	9	male	23.0 ± 0.8	22.6 ± 0.4	4.9± 0.1	-	NA	NA	4.9 ± 0.04	-	NA	NA
Vosselman et al. [40]	2012	RCT	10	male	22.5 ± 2.5	21.6 ± 1.6	4.9 ± 0.4	-	0.7 ± 0.2	324 ± 84	4.7 ± 0.4	-	0.7 ± 0.2	637 ± 398
Vosselman et al. [41]	2015	Comparative study	12	male	23.0 ± 3.3	21.8 ± 1.9	5.0 ± 0.3	66.5 ± 35.7	0.9 ± 0.6	567 ± 217	5.1 ± 0.4	85.4 ± 44.1	0.9 ± 0.5	671 ± 196
Din et al. [42]	2016	RCT	7	5/2	36.0 ± 11.0	25.5 ± 3.3	5.2 ± 0.3	49.7 ± 56.7	0.8 ± 0.3	NA	5.2 ± 0.2	33.6 ± 19.6	1.1 ± 0.4	NA
Chondronikola et al. [43]	2016	Crossover, non-RCT	16	male	47.8 ± 16.0	30.3 ± 2.1	NA	NA	1.5 ± 0.9	430 ± 220	NA	NA	1.6 ± 1.1	630 ± 280
Chondronikola et al. [44]	2017	Crossover, non-RCT	18	male	47.6 ± 17.8	29.7 ± 4.9	NA	36.6 ± 27.3	NA	NA	NA	32.9 ± 19.6	NA	NA
Mean ± SD	-	-	-	-	31.8 ± 11.0	24.9 ± 3.4	-	-	-	-	-	-	-	-

DATA are mean ± SD. (-): unconverted values: after converting these data from the presented units in the papers to the chosen unit in this review, these values were not in the biological range. Therefore, they were excluded. For more details, see Section 3.3. Abbreviations: not assessed in respective studies (NA); randomized controlled study (RCT); body mass index (BMI); triglycerides (TGs); free fatty acids (FFAs); cold exposure (CE); thermoneutrality (TN).

## Data Availability

Dataset available on request from the authors.

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
