# Peer review of "Metabolic Effects of Brown Adipose Tissue Activity Due to Cold Exposure in Humans: A Systematic Review and Meta-Analysis of RCTs and Non-RCTs"

_biomedicines, 2024, doi:10.3390/biomedicines12030537_

Round 1
Reviewer 1 Report
Comments and Suggestions for Authors In the manuscript entitled “Metabolic effects of brown adipose tissue activity due to cold exposure in humans: A systematic review and meta-analysis of RCTs and non-RCTs,” the author reviewed and assessed the metabolic effects of CE-induced BAT activation in healthy humans, examining changes in glucose and lipid metabolism compared to thermoneutrality (TN). I have some comments shown below to improve the clarity of this manuscript. 1. Consider incorporating the influence of age, gender, body weight, and the duration of cold exposure into the analysis to provide a more comprehensive understanding of the results. 2. It would be better to include references that support the use of the temperature range of 20-25 °C as conditions for thermoneutrality. 3. It would be better to correct “Error! Reference source not found” throughout the manuscript. 4. It would be better to include the results in lines 190 to 192 in Table 1.Author Response
Please see the attachment.

Reviewer 2 Report
Comments and Suggestions for Authors
The review is balanced and provides technical insight into the assessment of thermogenic fat in human. To have the whole picture, it would be important to mention a few points in the introduction:
1. Cold exposure is a key theme in this review. Thermogenic fat development is induced by various signals, other than cold (see in comment 2.). Moreover, preserving core body temperature is dependent on subcutaneous white fat depot thickness in human, instead of having thermogenic (brown, beige, brite) adipose tissue. Indeed, human populations living under arctic environment do not develop excess thermogenic fat. The thermogenic fat has a role only in warming up blood in large arteries and blood perfused organs (e.g. kidneys) in the adult human. It is important to note that in human the thermogenic fat rather limited impact on body temperature control (unlike in mice). Also, brown/beige fat is more relevant in the newborn than in the adult human. See here: (Sinclair 1978).
2. In the term human fetus and the newborn human, the core body temperature maintenance needs thermogenic subcutaneous fat, unlike in adults. The development of thermogenic fat is not a response to hypothermia, unlike in adults. Instead, maternal cues (FFAs and alkylglycerols) delivered through milk (Yu, Dilbaz et al. 2019, Pena-Leon, Folgueira et al. 2022) and microbial products of the gut microbiota (Tsukada, Okamatsu-Ogura et al. 2023) trigger thermogenic (beige/brown) fat development after birth. These signals stimulate fat thermogenesis through beta adrenergic signaling (Pena-Leon, Folgueira et al. 2022) and locally produced IL-6 (Yu, Dilbaz et al. 2019), independent of cold exposure. In human neonates breastfeeding is necessary for beige fat development, and this is again, independent of cold exposure (Yu, Dilbaz et al. 2019, Hoang, Sasi-Szabó et al. 2022, Gyurina, Yarmak et al. 2023).
3. Altogether, please address the above points in the introduction.
4. It would be really a good addition to discuss how can we assess thermogenic fat in infants and children, where in vivo imaging is limited (lack of cooperation and impossibility of immobilization) or even harmful (radiation exposure).
5. And lastly, in the summary section, please address the clinical impact of measuring brown fat mass in human, since the loss of thermogenic fat is an early indicator of obesity, hence brown fat volume can aid diagnosis or prognosis in pre-obesity and obesity (Ikeda, Maretich et al. 2018, Lim, Park et al. 2020, Gyurina, Yarmak et al. 2023).
References:
Gyurina, K., M. Yarmak, L. Sasi-Szabó, S. Molnár, G. Méhes and T. Röszer (2023). "Loss of Uncoupling Protein 1 Expression in the Subcutaneous Adipose Tissue Predicts Childhood Obesity." International Journal of Molecular Sciences 24(23): 16706.
Hoang, A. C., L. Sasi-Szabó, T. Pál, T. Szabó, V. Diedrich, A. Herwig, K. Landgraf, A. Körner and T. Röszer (2022). "Mitochondrial RNA stimulates beige adipocyte development in young mice." Nature Metabolism 4(12): 1684-1696.
Ikeda, K., P. Maretich and S. Kajimura (2018). "The Common and Distinct Features of Brown and Beige Adipocytes." Trends in Endocrinology & Metabolism 29(3): 191-200.
Lim, J., H. S. Park, J. Kim, Y. J. Jang, J.-H. Kim, Y. Lee and Y. Heo (2020). "Depot-specific UCP1 expression in human white adipose tissue and its association with obesity-related markers." International Journal of Obesity 44(3): 697-706.
Pena-Leon, V., C. Folgueira, S. Barja-Fernández, R. Pérez-Lois, N. Da Silva Lima, M. Martin, V. Heras, S. Martinez-Martinez, P. Valero, C. Iglesias, M. Duquenne, O. Al-Massadi, D. Beiroa, Y. Souto, M. Fidalgo, R. Sowmyalakshmi, D. Guallar, J. Cunarro, C. Castelao, A. Senra, P. González-Saenz, R. Vázquez-Cobela, R. Leis, G. Sabio, H. Mueller-Fielitz, M. Schwaninger, M. López, S. Tovar, F. F. Casanueva, E. Valjent, C. Diéguez, V. Prevot, R. Nogueiras and L. M. Seoane (2022). "Prolonged breastfeeding protects from obesity by hypothalamic action of hepatic FGF21." Nature Metabolism 4(7): 901-917.
Sinclair, J. C. (1978). Temperature Regulation And Energy Metabolism In The Newborn (Monographs in Neonatalogy), Grune & Stratton, Florence, Kentucky, U.S.A.
Tsukada, A., Y. Okamatsu-Ogura, E. Futagawa, Y. Habu, N. Takahashi, M. Kato-Suzuki, Y. Kato, S. Ishizuka, K. Sonoyama and K. Kimura (2023). "White adipose tissue undergoes browning during preweaning period in association with microbiota formation in mice." iScience 26(7).
Yu, H., S. Dilbaz, J. Coßmann, A. C. Hoang, V. Diedrich, A. Herwig, A. Harauma, Y. Hoshi, T. Moriguchi, K. Landgraf, A. Körner, C. Lucas, S. Brodesser, L. Balogh, J. Thuróczy, G. Karemore, M. S. Kuefner, E. A. Park, C. Rapp, J. B. Travers and T. Röszer (2019). "Breast milk alkylglycerols sustain beige adipocytes through adipose tissue macrophages." The Journal of Clinical Investigation 129(6): 2485-2499.
Reviewer 3 Report
Comments and Suggestions for Authors
The authors conducted a meta-analysis on the metabolic effects of brown adipose tissue activity in human by cold exposure. Considering the epidemics of obesity and the emerging therapeutic potential of brown adipose tissue in combating obesity and its-associated diseases, this study provides a timely and comprehensive view in the field and will arouse a lot of interest. Generally speaking the manuscript is well written and easy to follow, although some types and mistakes are to be corrected (see below). The number of studies to be included is low but feasible explanation has been discussed in the manuscript.
Major points:
1. Figure 5 is a duplicate of Figure 4. Please replace.
2. There are underlie tiles under the words “standardized” and “difference” in Figure 2-5. Please amend.
3. Since the correct Figure 5 is not shown, I cannot tell whether the description and the conclusion for the FFA part are feasible or not.
4. In the second paragraph of the discussion, a number of studies have been listed and described. But a more insightful conclusion from these studies shall be drawn, and possible reasons leading to either the consistency or inconsistency between different studies shall be discussed.
Minor issues:
1. Line 74, what does “non-shivering” mean here in front of CE? Is it a typo?
2. line 121, “For more details 0.”? This is a typo.
3. line 140 and 129,” confidence intervals (Cl)” were defined twice. And they should be CI, not Cl.
Round 2
Reviewer 3 Report
Comments and Suggestions for Authors
All the concerns have been addressed.